# ADAPTIVE INPUT REPRESENTATIONS FOR NEURAL LANGUAGE MODELING

**Alexei Baevski & Michael Auli**
Facebook AI Research,
Menlo Park, CA, USA

## ABSTRACT

We introduce adaptive input representations for neural language modeling which extend the adaptive softmax of Grave et al. (2017) to input representations of variable capacity. There are several choices on how to factorize the input and output layers, and whether to model words, characters or sub-word units. We perform a systematic comparison of popular choices for a self-attentional architecture. Our experiments show that models equipped with adaptive embeddings are more than twice as fast to train than the popular character input CNN while having a lower number of parameters. On the WIKITEXT-103 benchmark we achieve 18.7 perplexity, an improvement of 10.5 perplexity compared to the previously best published result and on the BILLION WORD benchmark, we achieve 23.02 perplexity.[1]

## 1    INTRODUCTION

Language modeling is a basic task in natural language processing, with many applications such as speech recognition (Arisoy et al., 2012) and statistical machine translation (Schwenk et al., 2012; Vaswani et al., 2013; Baltescu & Blunsom, 2015). Recently, much progress has been made by neural methods (Bengio et al., 2003; Mikolov et al., 2010) based on LSTMs (Józefowicz et al., 2016), gated convolutional networks (Dauphin et al., 2017) and self-attentional networks (Al-Rfou et al., 2018).

There are different choices for the basic unit we wish to model, including full words (Bengio et al., 2003), characters for the input (Kim et al., 2016), or also the output (Merity et al., 2018) as well as sub-words (Buckman & Neubig, 2018; Mielke & Eisner, 2018). Word-based models are particularly challenging since computing probabilities for all 800K words of the BILLION WORD benchmark is still a substantial part of the overall computation (Chen et al., 2016).

A popular approach to lower the computational burden is to structure the output vocabulary so that not all probabilities need to be computed. The hierarchical softmax does this by introducing latent variables or clusters to simplify normalization (Goodman, 2001; Morin & Bengio, 2005; Mikolov et al., 2011). This has been further improved by the *adaptive softmax* which introduces a variable capacity scheme for output word embeddings, assigning more parameters to frequent words and fewer parameters to rare words (Grave et al., 2017).

In this paper, we introduce *adaptive input embeddings* which extend the adaptive softmax to input word representations. This factorization assigns more capacity to frequent words and reduces the capacity for less frequent words with the benefit of reducing overfitting to rare words. For a competitive setup on the BILLION WORD benchmark, adaptive input embeddings reduce the number of parameters in the input and output layers by 23% while achieving higher accuracy over fixed size embeddings. When the adaptive input representations are tied with an adaptive softmax in the output, then the number of parameters is reduced by a total of 61%.

Our experiments compare models based on word inputs, character inputs, as well as sub-word units using a self-attention architecture (Vaswani et al., 2017). We show that models with adaptive word representations can outperform very strong character-based models while training more than twice as fast. We also substantially improve adaptive softmax by introducing additional dropout regularization in the tail projection. On the WIKITEXT-103 benchmark we achieve a perplexity of 18.7, a

---

[1]Code and pre-trained models available at `http://github.com/pytorch/fairseq`

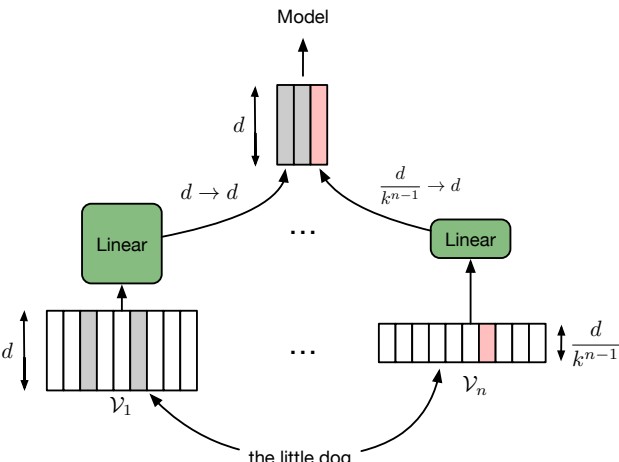

Figure 1: Illustration of adaptive input representations. Words are assigned to clusters $\mathcal{V}_i$ based on their frequency which determines the size of the representations. Embeddings are projected to a common dimension $d$ before being fed to the model.

reduction of 10.5 perplexity over the previously best published result. On the larger BILLION WORD benchmark our best model with adaptive input embeddings achieves 23.02 perplexity, a reduction of nearly 5 perplexity over the next best previously published result.

## 2   RELATED WORK

Adaptive word representations are inspired by the adaptive softmax work Grave et al. (2017) which first described a GPU friendly way to construct a hierarchical softmax and showed that it performs very competitively compared to a full softmax, while offering significantly faster speed and a lower memory footprint.

Merity et al. (2018) use a modified version of adaptive softmax which does not reduce the dimensionality of less frequent words in order to be able to share output embeddings with the input. This setup is akin to a hierarchical softmax with tied weights. We show that variable-sized input embeddings can perform better than fixed sized embeddings. Furthermore, this also enables weight sharing with an adaptive softmax output layer.

Merity et al. (2018) evaluates both character-based and word-based factorizations but does not directly compare them to each other. We perform a direct comparison of word-based and character-based input vocabularies and also compare to a sub-word factorization for both the input and output. Recently, Al-Rfou et al. (2018) demonstrated that self-attentional models can perform very well on language modeling tasks where the input and output is both characters. We also consider word-based benchmarks.

## 3   ADAPTIVE INPUT REPRESENTATIONS

The adaptive softmax exploits the fact that the distribution of word types in natural language follows a Zipfian distribution in order to improve the computation of the output probabilities. We apply the same intuition for input word embeddings with the motivation to reduce the number of parameters which frees up capacity for other parts of the model.

We define a number of clusters that partitions the frequency ordered vocabulary $\mathcal{V} = \mathcal{V}_1 \cup \mathcal{V}_2, \ldots, \mathcal{V}_{n-1} \cup \mathcal{V}_n$ such that $\mathcal{V}_i \cap \mathcal{V}_j = \emptyset$ for $\forall i, j$, and $i \neq j$, where $\mathcal{V}_1$ contains the most frequent words and $\mathcal{V}_n$ the least frequent words. We will refer to $\mathcal{V}_1$ as the *head* and to any subsequent clusters loosely as *tail*. We reduce the capacity for each cluster by a factor of $k$. That is, if words in $\mathcal{V}_1$ have dimension $d$, then words in $\mathcal{V}_n$ have dimension $\frac{d}{k^{n-1}}$. We typically set $k = 4$ following Grave et al. (2017).

Next, we add linear projections $W_1 \in \mathbb{R}^{d \times d}, \ldots, W_n \in \mathbb{R}^{d/k^{n-1} \times d}$ to map the embeddings of each cluster to dimension $d$ so that the concatenated output of the adaptive input embedding layer can be easily used by the subsequent model (Figure 1). We also project $\mathcal{V}_1$ which already has dimension $d$.

When presented with a number of input words, the adaptive input embedding layer partitions the words into the various clusters, performs separate lookups in the embedding tables and then projects to dimension $d$, followed by concatenating the embeddings in the original order.

**Weight sharing.** When the output layer is an adaptive softmax with the same partition of $\mathcal{V}$, $d$, and $k$ as the adaptive input layer, then we can tie the weights (Inan et al., 2016; Press & Wolf, 2017). This further reduces the number of parameters and can simultaneously improve performance (§5). We can share both the parameters for the actual words as well as the projections $W_1, \ldots, W_n$ Sharing the word embeddings is straightforward except for the head where the adaptive softmax has $n-1$ additional embeddings for the remaining clusters which are not shared with the input.

We share all projections, except for the head projection which is not available in the adaptive softmax since the model output is directly multiplied with the output word embeddings for the head band. Performance decreased when we added a head projection to the adaptive softmax in the output, regardless of when it was shared or not. Sharing both the word embeddings as well as the projections performed very well on WIKITEXT-103 but on BILLION WORD we only share the word embeddings as we found that this performed better on the validation set.

## 4 EXPERIMENTAL SETUP

### 4.1 MODEL

We follow most of the architectural choices described in Vaswani et al. (2017) but use only a decoder network. We add sinusoidal position embeddings to the input layer and stack $N = 16$ blocks for both BILLION WORD and WIKITEXT-103. Each block contains two sub-blocks: the first is a multi-head self-attention module with $H = 16$ heads. The second sub-block is a feed-forward module (FFN) of the form $ReLU(\boldsymbol{W}_1 \boldsymbol{X} + b_1)\boldsymbol{W}_2 + b_2$ where $\boldsymbol{W}_1 \in \mathbb{R}^{e \times e_{ff}}$, $\boldsymbol{W}_1 \in \mathbb{R}^{e_{ff} \times e}$ and $e = 1024$, $e_{ff} = 4096$ unless otherwise stated. Different to Vaswani et al. (2017) we apply layer normalization before the self-attention and FFN blocks instead of after, as we find it leads to more effective training. Sub-blocks are surrounded by a residual connection (He et al., 2015).

We use a dropout rate of 0.1 and attention dropout of 0.1 for BILLION WORD models, and increase regularization for WIKITEXT-103 by using dropout 0.3, and 0.1 ReLU dropout as well as attention dropout 0.1. We use the same hyperparameters for all models trained on the same dataset in order to enable a like for like comparison. When the dimensionality of the input or output layer differs from $e$, then we add a simple linear projection with no bias.

### 4.2 DATASETS

We experiment on the BILLION WORD benchmark and WIKITEXT-103. BILLION WORD contains 768M word tokens and has a vocabulary of about 800K word types, which corresponds to words with more than 3 occurrences in the training set (Chelba et al., 2013).

The training data of WIKITEXT-103 comprises about 100M tokens and a vocabulary of around 260K, corresponding to types with more than 3 occurrences in the training data (Merity et al., 2016). The dataset is composed of shuffled Wikipedia articles where the context carries across sentences.

### 4.3 BATCHING

For BILLION WORD we batch individual sentences since the corpus does not contain document structure. For WIKITEXT-103 we partition the training data into blocks of 512 contiguous tokens ignoring document boundaries. Evaluation is the same except that we require blocks to contain complete sentences totaling up to 512 tokens.[2]

---

[2] Respecting document boundaries may lead to better results and we leave this to future work.

We limit the number of tokens per GPU to a maximum threshold $B$ per GPU. That is, we add examples of similar length until we reach this threshold. When we train on multiple GPUs, each GPU processes $B$ tokens using the same model parameters. This increases the effective batch size to the product of the number of GPUs and $B$. For BILLION WORD models we use $B = 2048$ and typically train on 32 GPUs, giving an effective batch size of 65K tokens. The smaller vocabulary of WIKITEXT-103 enables increasing $B$ to 4096 and we train on 8 GPUs. We found that large batch training is beneficial for this dataset and we therefore accumulate gradient updates over two batches before committing a parameter update (Ott et al., 2018a). This gives an effective batch size of 65K tokens for WIKITEXT-103.

## 4.4 INPUT AND OUTPUT LAYER HYPERPARAMETERS

**Embedding sizes.** For fixed size word input layers and softmax output layers we generally use embeddings of size 512 for WIKITEXT-103. When we use an adaptive softmax in the output and fixed size word embeddings for the input, then we use dimension 256 for the input embeddings for BILLION WORD and 64 for WIKITEXT-103. We tuned this choice on the validation set (Appendix A). BPE inputs and outputs have embeddings of size 1024.

**Character CNN.** We model character inputs by convolving the representations of all characters in a word following Kim et al. (2015) which applies several filters, then max pooling, a number of highway layers and a projection. Character embeddings have size 128 and we apply seven filters of size 1x128, 2x256, 3x384, 4x512, 5x512, 6x512, 7x512, where 3x128 indicates a filter processing three characters that outputs 128 features. We use a single highway layer for WIKITEXT-103, and two for BILLION WORD. We do not add start of word and end of word markers as they did not improve validation accuracy. We train on the same pre-processed data as the other models, with unknown tokens in both the inputs and outputs.

**Adaptive input representations and adaptive softmax.** We use an adaptive softmax output layer to train models with large word-based vocabularies. For adaptive word inputs and adaptive softmax, we use embeddings of size $d = 1024$ for the head and reduce the size of subsequent clusters by a factor of $k = 4$. For WIKITEXT-103, we have three bands of size 20K (d=1024), 40K (d=256) and 200K (d=64). For BILLION WORD the bands are 60K (d=1024), 100K (d=256), and 640K (d=64).

**Sub-word models.** We learn a byte-pair encoding (BPE) of 32K codes on the training data of each benchmark (Sennrich et al., 2016). After applying the code to the training data we obtain a vocabulary of 33,337 tokens for WIKITEXT-103 and 32,347 tokens for BILLION WORD. BPE input/output embeddings have size 1024. The final evaluation is in terms word-level perplexity to be comparable to other models. The probability of a word is the product of the sub-word units.

## 4.5 OPTIMIZATION

Different to Vaswani et al. (2017) we use Nesterov's accelerated gradient method (Sutskever et al., 2013) with a momentum of 0.99 and we renormalize gradients if their norm exceeds 0.1 (Pascanu et al., 2013). The learning rate is linearly warmed up from $10^{-7}$ to 1 for 16K steps and then annealed using a cosine learning rate schedule with $C$ cycles (Loshchilov & Hutter, 2016). Each cycle runs for twice the number of updates than the previous cycle and we lower the maximum and minimum learning rates by a rate $M$ compared to the previous cycle. The initial minimum learning rate is $10^{-5}$ and the maximum is 1.

BILLION WORD models train for a total of 975K updates over $C = 3$ cycles, the first cycle takes 137K steps, and we set $M = 0.6$. The WIKITEXT-103 models train for 286K steps over $C = 4$ cycles, the first cycle takes 18K setps and we set $M = 0.75$. We run experiments on DGX-1 machines with 8 NVIDIA V100 GPUs and machines are interconnected by Infiniband. We also use the NCCL2 library and the torch.distributed package for inter-GPU communication. We train models with 16-bit floating point precision, following Ott et al. (2018b).

| | Test | Train Time (hours) | Parameters |
|---|---|---|---|
| Dauphin et al. (2017) | 31.9 | - | 428M |
| Józefowicz et al. (2016) | 30.0 | - | 1,040M |
| Shazeer et al. (2017) | 28.0 | - | 4,371M[†] |
| Char-CNN | 25.88 | 79 | 366M |
| Adaptive inputs | 25.22 | 55 | 331M |
| Adaptive inputs (large) | 23.91 | 72 | 465M |
| Adaptive inputs (very large) | **23.02** | 145 | 1026M |
| 10 LSTMs + SNM10-SKIP (Shazeer et al., 2016) | 23.7 | - | - |

Table 1: Test perplexity on BILLION WORD. Adaptive inputs share parameters with an adaptive softmax. Training times of Char-CNN and Adaptive input models are measured when training with 64 GPUs.

[†] does not include embedding and softmax layers

| | Test | Train Time (hours) | Parameters |
|---|---|---|---|
| Grave et al. (2016) | 40.8 | - | |
| Dauphin et al. (2017) | 37.2 | - | 229M |
| Merity et al. (2018) | 33.0 | - | 151M |
| Rae et al. (2018) | 29.2 | - | |
| Adaptive inputs | **18.7** | 67 | 247M |

Table 2: Test perplexity on WIKITEXT-103 (cf. Table 1). Training time is based on 8 GPUs.

## 5 EXPERIMENTS AND RESULTS

### 5.1 MAIN RESULTS

For the main results on BILLION WORD, we doubled the batch size by training on 64 GPUs instead of 32 GPUs. We also consider two larger setups, one where we added four more blocks ($N = 20$) and increased the FFN dimension to $e_{ff} = 6144$ (large), and another where we add another four blocks ($N = 24$) with $e_{ff} = 8192$ and $e = 1536$ (very large). All other settings follow §4.4 and all models were trained for the same number of steps.

Table 1 compares our models to previous work on BILLION WORD. The adaptive input model outperforms the best previously reported result at an order of magnitude fewer parameters. Our large model performs nearly as well as an ensemble of over ten models and achieves a new state of the art of 24.14 perplexity. Our very large model performs as well as an ensemble of over ten models and achieves 23.02 perplexity. The Char-CNN model performs 0.6 PPL worse than the standard adaptive input model even though it trained for over 40% longer.

Table 2 shows our result on WIKITEXT-103 where adaptive inputs achieve 18.7 perplexity. For this result only, we partition the training data into blocks of 3072 contiguous tokens instead of 512 tokens as for other experiments. During evaluation we require blocks to contain complete sentences totaling up to 3072 tokens of which the first 2560 tokens serve as context to score the last 512 tokens; we take care to score all tokens in the test and validation sets. We motivate this choice in §5.3.

### 5.2 COMPARISON OF INPUT AND OUTPUT LAYER FACTORIZATIONS

Next, we perform a systematic comparison of different input and output layer factorizations. We consider a word-based setup with fixed size word input embeddings and a standard word softmax (SM) where embeddings have either dimension 512 (WIKITEXT-103) or 64 (BILLION WORD). We consider tying the input and output embeddings (SM-T). Instead of words, we try less sparse sub-

| | Input | Output | Valid | Test | Train Time (hours) | Params |
|---|---|---|---|---|---|---|
| SM | Embedding | Softmax | 23.87 | 24.92 | 57* | 476.8M |
| BPE | BPE Embedding | BPE Softmax | 23.13 | 24.25 | 30 | 270M |
| BPE-T | BPE Embedding | BPE Softmax (tied) | 22.46 | 23.45 | 30 | 235.7M |
| SM-T | Embedding | Softmax (tied) | 22.63 | 23.38 | 56* | 339.7M |
| ASM | Embedding | Adaptive | 21.23 | 22.18 | 35 | 263.1M |
| CNN | Char-CNN | Adaptive | 20.86 | 21.79 | 70 | 266.3M |
| ADP | Adaptive | Adaptive | 20.95 | 21.74 | 34 | 291.3M |
| ADP-T | Adaptive | Adaptive (tied) | **19.79** | **20.51** | 30 | 246.9M |

Table 3: Test perplexity on WIKITEXT-103 for various input and output layer factorizations. Training speed was measured on a single 8-GPU machine. (*) indicates a modified training regime because of large memory requirements: the maximum number of tokens per GPU was lowered to 1024 from 4096 but the same number of updates were performed by processing four batches before committing a weight update.

| | Input | Output | Valid | Test | Train time (hours) | Params |
|---|---|---|---|---|---|---|
| BPE-T | BPE Embedding | BPE Softmax (shared) | 27.44 | 27.51 | 34 | 234.7M |
| BPE | BPE Embedding | BPE Softmax | 27.02 | 27.13 | 35 | 267.8M |
| ASM | Embedding | Adaptive | 26.97 | 27.06 | 62 | 532.8M |
| CNN | Char-CNN | Adaptive | 26.13 | 26.25 | 92 | 365.8M |
| ADP | Adaptive | Adaptive | 26.38 | 26.49 | 65 | 458.4M |
| ADP-T | Adaptive | Adaptive (shared) | **25.51** | **25.58** | 43 | 330.8M |

Table 4: Test perplexity on BILLION WORD. Training speed measured on four 8-GPU machines.

word units, both in the input and output, with embeddings of size 1024 (BPE) and shared weights (BPE-T). Next, we consider replacing the fixed size output representations by an adaptive softmax (ASM) and characters as input (CNN). Finally, we use both adaptive input word representations as well as an adaptive softmax (ADP) and a tied version (ADP-T). All models use the same self-attention architecture described in §4.1.

Table 3 shows results when training all configurations for the same number of updates. Adaptive input representations with tied input and output layers (ADP-T) achieve the highest accuracy at the same speed as the BPE models which have a very small vocabulary (33K versus 260K). CNN is 1 perplexity worse than ADP-T and requires well over twice the training time. It is the slowest approach, even though it has a fast adaptive softmax in the output. Fixed word embeddings perform least well (SM). Sub-word units are fast to train and perform better than word models with fixed sized embeddings. ASM improves over SM and greatly speeds up training. For ASM, we found that reducing the dimension of the input word embeddings to 64 on WIKITEXT-103 results in better accuracy (Appendix A).

Table 4 shows that adaptive input representations perform equally well on BILLION WORD compared to other factorizations. ADP-T is 34% faster than ADP because there are fewer parameters to update. Similar to before, ADP-T trains more than twice as fast as CNN at higher accuracy, however, the accuracy gap is narrower than for WIKITEXT-103. Regularization is more important on WIKITEXT-103 while models for BILLION WORD benefit from additional capacity. Because of this we used input word embeddings of size 256 for ASM.

We also trained CNN without replacing input words outside the vocabulary by an unknown symbol, however, this only improved validation perplexity by 0.16.

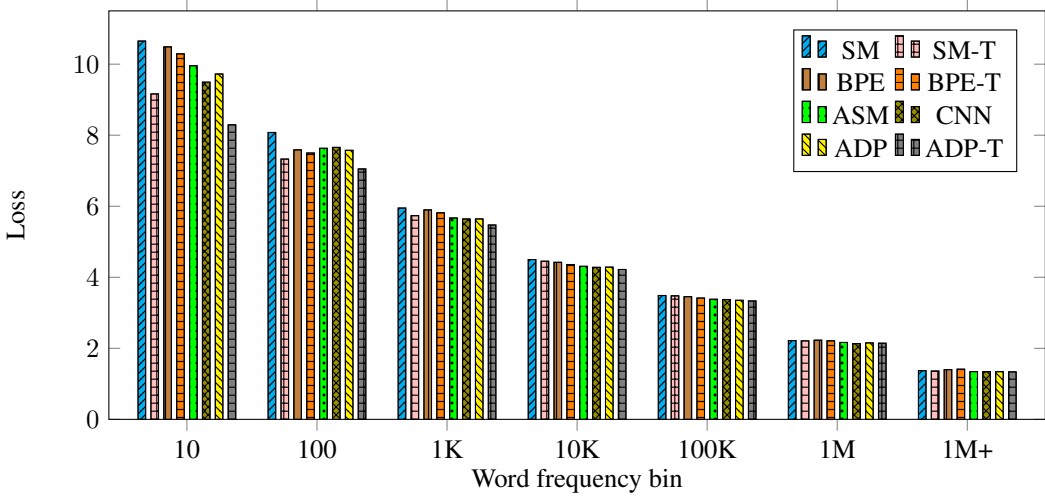

Figure 2: Loss of models binned by word frequency on the test set of WIKITEXT-103. Bins are not cumulative.

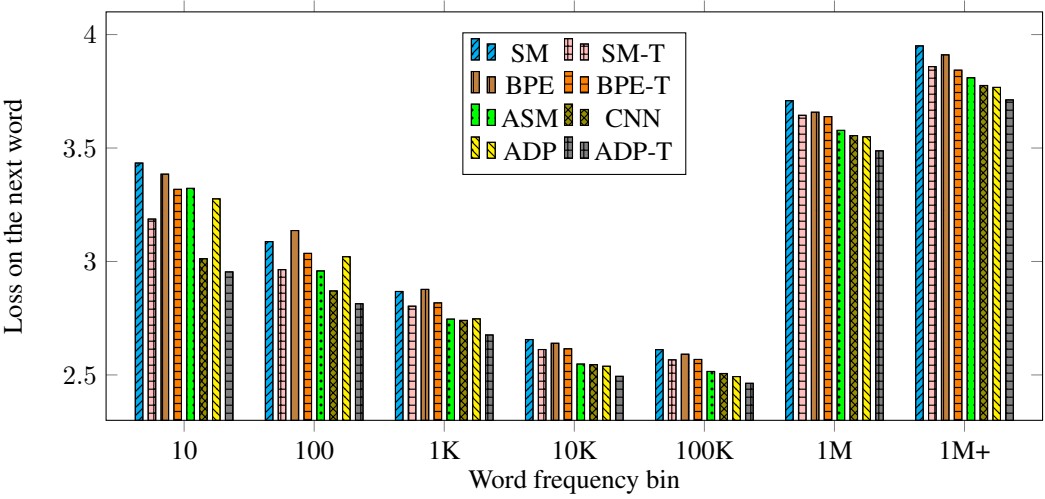

Figure 3: Loss of models when binning by the frequency of the *previous* word measured on WIKITEXT-103 (cf. Figure 2).

### 5.3 ANALYSIS

Next, we turn to the question of how well models perform on rare words compared to frequent words. We compute the average loss for each word in the test set and group words by frequency.

Figure 2 shows results on WIKITEXT-103. Tying weights helps all models on rare words, likely because of regularization effects. Fixed size word embeddings with a word softmax (SM and SM-T) do not perform well on rare words. This is likely due to underfitting on common words and we use the largest possible embedding size we could fit on 16GB GPU cards given our batch size (more experimentation in Appendix A). BPE and BPE-T perform poorly on rare words because probabilities are a product of several sub-word units. ADP-T performs best across all frequency ranges. Figure 3 bins the loss by the frequency of the *previous* word and shows that CNN does well when it has rare words in the context, however, ADP-T does best across all bins.

Figure 4 shows an equivalent analysis for BILLION WORD. The largest differences between models is on rare words. CNN performs best on very rare words but is outperformed by ADP in all other settings. Similar to WIKITEXT-103, BPE and BPE-T perform poorly on rare words. Further

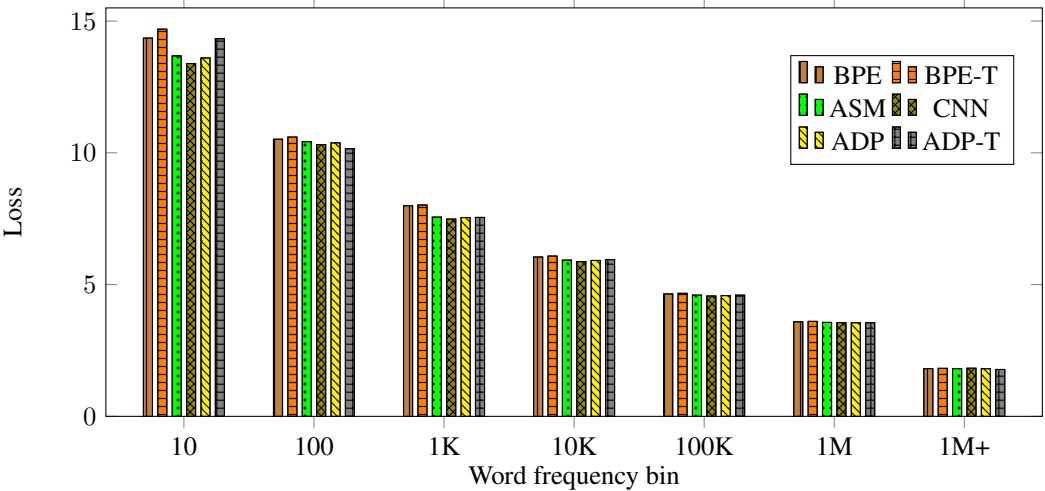

Figure 4: Loss of models when binning by word frequency on the test set of BILLION WORD. Bins are not cumulative.

| Train block size | Inference context size | Validation perplexity | Test perplexity |
|---|---|---|---|
| 512 | 0 | 19.79 | 20.51 |
| 512 | 480 | 18.35 | 19.03 |
| 2048 | 0 | 18.96 | 19.53 |
| 2048 | 1536 | 18.23 | 18.88 |
| 3072 | 0 | 18.63 | 19.34 |
| 3072 | 2560 | 17.97 | 18.70 |

Table 5: Perplexity on WIKITEXT-103 with different context sizes during training and inference. Training block size is the number of consecutive tokens considered during training. Inference context is the number of tokens provided at evaluation before scoring tokens.

analysis (Appendix 5.3) binning the loss by the frequency of the previous word shows that weight sharing also helps for BILLION WORD and that CNN does very well on rare words for BILLION WORD compared to other models.

Table 5 shows the importance of context size for WIKITEXT-103. Training block size is the number of consecutive tokens that are considered at once during training. Inference context is the number of tokens that are provided at evaluation before any tokens are scored. Simply increasing the training block size from 512 to 3072 results in an improvement of nearly 1.2 perplexity with no inference context window. Increasing the context size at inference time results in an improvement of 0.6 perplexity for the largest training block size.

## 5.4 ADAPTIVE SOFTMAX VS. FULL SOFTMAX

We also found that adaptive softmax can benefit from additional regularization of rare words. Adaptive softmax first projects the model output to the dimension of a particular cluster and then computes a dot product with the respective word embeddings. We add dropout to the output of the first projection for all clusters, except for the head. This change enables the adaptive softmax to outperform a standard softmax over fixed size output word embeddings on WIKITEXT-103 (Table 6).

However, we found that adding dropout in this way is not helpful for larger datasets such as BILLION WORD. Unfortunately, a standard softmax over 800K words is not tractable and we were unable to make a comparison. It may be possible to achieve better results by tuning dropout for each band of the tail and we leave this for future work.

|  | Tail dropout | Validation perplexity |
|---|---|---|
| Softmax (SM) | N/A | 23.87 |
| Adaptive (ADP) | 0.0 | 24.74 |
| Adaptive (ADP) | 0.2 | 21.23 |

Table 6: Perplexity on WIKITEXT-103 when regularizing rare words in adaptive softmax.

## 6 CONCLUSION

Adaptive input embeddings vary the size of input word embeddings which can improve accuracy while drastically reducing the number of model parameters. When sharing parameters with an adaptive softmax, the number of parameters can be further reduced which improves training speed. We presented a comparison between different input and output layer factorizations including word inputs, character inputs and sub-word units in both the input and output.

Our experiments show that models with adaptive input embeddings train faster compared to character input CNNs while achieving higher accuracy. We achieve new state of the art results on WIKITEXT-103 and BILLION WORD. In future work, we will apply variable sized input embeddings to other tasks.

ACKNOWLEDGMENTS

We thank Tom Bosc for fruitful comments and suggestions.

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

SUPPLEMENTARY MATERIAL

## A    ADDITIONAL EXPERIMENTS ON WIKITEXT-103

This appendix shows various ablation. Table 7 shows that reducing the capacity of fixed size word input embddings is beneficial on WIKITEXT-103. The next set of results in Table 7 shows results for various settings of the SM and SM-T models. We also experimented with sharing the head projection but found this to perform less well than not sharing it. Finally, Table 8 shows various band sizes for adaptive input word embbedings.

We also show the performance of Merity et al. (2018) who use an adaptive softmax with equally sized word representations and share the input and output embeddings (no dim reduction, tied).

| Input | Output | Dropout | Valid PPL | Parameters |
|---|---|---|---|---|
| 256d Embedding | Adaptive | 0.3 | 23.39 | 314.7M |
| 128d Embedding | Adaptive | 0.3 | 21.51 | 280.3M |
| 64d Embedding | Adaptive | 0.3 | 21.23 | 263.1M |
| 32d Embedding | Adaptive | 0.3 | 21.78 | 254.5M |
| 512d Embedding | 512d Softmax (tied) | 0.3 | 22.63 | 339.7M |
| 512d Embedding | 512d Softmax (tied) | 0.4 | 28.31 | 339.7M |
| 512d Embedding | 512d Softmax | 0.3 | 23.87 | 476.8 |
| 512d Embedding | 512d Softmax | 0.4 | 27.64 | 476.8 |
| 256d Embedding | 256d Softmax (tied) | 0.3 | 22.65 | 270.6M |
| 256d Embedding | 256d Softmax | 0.3 | 24.13 | 339.1M |
| 64d Embedding | 512d Softmax | 0.3 | 24.74 | 356.3M |
| Adaptive | Adaptive (tied emb, not proj) | 0.3 | 20.06 | 247.3M |
| Adaptive | Adaptive (tied emb/proj not head) | 0.3 | 19.79 | 246.9M |
| Adaptive | Adaptive (tied emb/proj + head) | 0.3 | 20.06 | 246.9M |
| 512d Embedding | 512d Softmax (tied) | 0.3 | 22.63 | 339.7M |
| 512d Embedding | 512d Adaptive (no dim reduction, tied) | 0.3 | 25.48 | 340.2M |

Table 7: Validation perplexity of our models on WIKITEXT-103.

| Softmax cutoff | Valid PPL |
|---|---|
| 20k/40k/200k | 19.79 |
| 20k/140k/100k | 20.26 |
| 20k/40k/60k/140k | 20.53 |
| 60k/100k/100k | 20.52 |
| 5k/155k/100k | 20.06 |
| 20k/40k/200k | 19.99 |
| 10k/60k/190k | 19.79 |

Table 8: Validation perplexity on WIKITEXT-103 with tied adaptive inputs & outputs. The bands signify the number of words belonging to each band. In every case, the first band has dimension 1024, the second band 256, the third band 64 and the fourth band (if it exists) 16.

## B    ANALYSIS

This appendix extends the analysis in §5.3 by showing a breakdown of the test loss when binning by the frequency of the previous word.

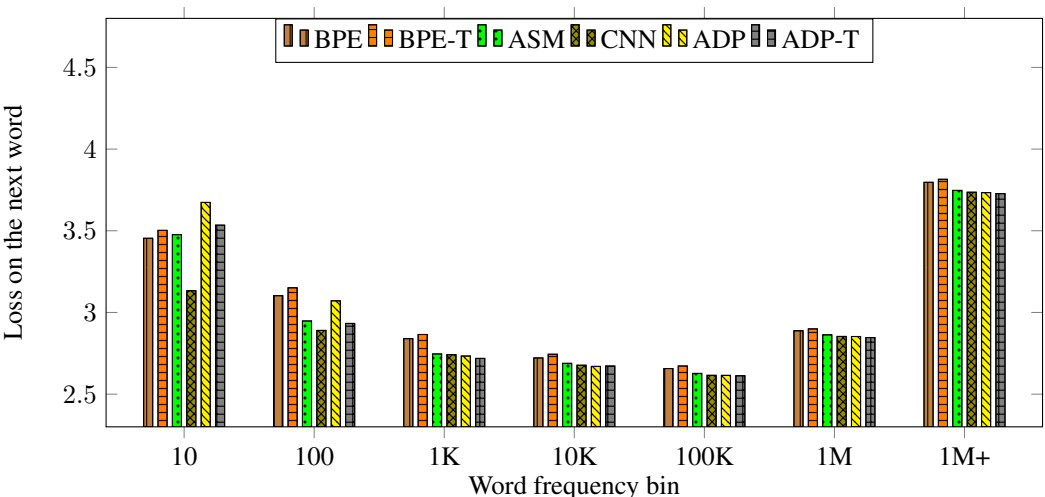

Figure 5: Loss of models when binning by the frequency of the *previous* word measured on BILLION WORD (cf. Figure 3).

