# OpenReview forum: "Adaptive Input Representations for Neural Language Modeling"
_ICLR.cc/2019/Conference_

### Official Review · AnonReviewer3 · 2018-11-01
**Reasonable increment from Grave et al. (2017)**

**Rating:** 8
**Confidence:** 4

**Review:**

This article presents experiments on medium- and large-scale language modeling when the ideas of adaptive softmax (Grave et al., 2017) are extended to input representations.

The article is well written and I find the contribution simple, but interesting. It is a reasonable and well supported increment from adaptive softmax of Grave et al. (2017).

My question is a bit philosophical: The only thing which I was concerned about when reading the paper is projection of the embeddings back to the d-dimensional space. I understand that for two matrices A and B we have rank(AB) <= min(rank(A), rank(B)), and we are not making the small-sized embeddings richer when backprojecting to R^d, but have you thought about how it would be possible to avoid this step and keep the original variable-size embeddings?

References
Joulin, A., Cissé, M., Grangier, D. and Jégou, H., 2017, July. Efficient softmax approximation for GPUs. In International Conference on Machine Learning (pp. 1302-1310).

---

> ### Author Response · Authors · 2018-11-20
> **Response to Reviewer #3**
>
> The primary goal of the projections is to project all embeddings into the model dimension d so that we can have variable sized embeddings. Our goal was not to make the model model expressive. Compared to the rest of the model, these projections add very little overhead compared to the rest of the model. Doing without them is an interesting future direction though!

---

### Official Review · AnonReviewer2 · 2018-11-02
**simple model architecture changed and extensive experiments**

**Rating:** 8
**Confidence:** 4

**Review:**

This paper introduced a new architecture for input embeddings of neural language models: adaptive input representation (ADP). ADP allowed a model builder to define a set of bands of input words with different frequency where frequent words have larger embedding size than the others. The embeddings of each band are then projected into the same size. This resulted in lowering the number of parameters.

Extensive experiments with the Transformer LM on WikiText-103 and Billion Word corpus showed that ADP achieved competitive perplexities. While tying weight with the output did not benefit the perplexity, it lowered the runtime significantly on Billion Word corpus. Further analyses showed that ADP gained performance across all word frequency ranges.

Overall, the paper was well-written and the experiments supported the claim. The paper was very clear on its contribution. The variable-size input of this paper was novel as far as I know. However, the method, particularly on the weight sharing, lacked a bit of important background on adaptive softmax. The weight sharing was also needed further investigation and experimental data on sharing different parts.

The experiments compared several models with different input levels (characters, BPE, and words). The perplexities of the proposed approach were competitive with the character model with an advantage on the training time. However, the runtimes were a bit strange. For example, ADP and ADP-T runtimes were very close on WikiText-103 dataset but very different on Billion Word corpus (Table 3 and 4). The runtime of ADP seemed to lose in term of scaling as well to BPE. Perhaps, the training time was an artifact of multi-GPU training.

Questions:
1. I am curious about what would you get if you use ADP on BPE vocab set?
2. How much of the perplexity reduction of 8.7 actually come from ADP instead of the transformer and optimization?

---

> ### Author Response · Authors · 2018-11-20
> **Response to Reviewer #2**
>
> We thank the reviewer for the comments!
>
> Q: “ADP and ADP-T runtimes were very close on WikiText-103 dataset but very different on Billion Word corpus (Table 3 and 4)”
> The differences in training time are due to the size of the models: Weight tying saves a lot more parameters for the Billion Word model due to the larger vocab compared to the WikiText-103 models which have a smaller vocab. On WikiText-103, tying saves 15% of parameters (Table 3, ADP vs ADP-T, 291M vs 247M) and training time is reduced by about 13%. On Billion Word, tying saves 27% of parameters (Table 4) and training time is reduced by about 34%. The slight discrepancy may be due to multi-machine training for Billion Word compared to the single machine setup for WikiText-103.
>
> Q1: "I am curious about what would you get if you use ADP on BPE vocab set?"
> We tried adaptive input embeddings with BPE but the results were worse than softmax. This is likely because 'rare' BPE units are in some sense not rare enough compared to a word vocabulary. In that case, the regularization effect of assigning less capacity to 'rare' BPE tokens through adaptive input embeddings is actually harmful.
>
> Q2: "How much of the perplexity reduction of 8.7 actually come from ADP instead of the transformer and optimization?"
> For WikiText-103 (Table 3) we measured 24.92 on test with a full softmax model (a 5.2 PPL improvement over the previous SOTA). This corresponds to a Transformer model including our tuned optimization scheme. Adding tied adaptive input embeddings (ADP-T) to this configuration reduces this perplexity to 20.51, which is another reduction of 4.4 PPL.

---

### Official Review · AnonReviewer1 · 2018-11-06
**Solid contribution to the language modeling literature**

**Rating:** 7
**Confidence:** 4

**Review:**

The authors extend an existing approach to adaptive softmax classifiers used for the output component of neural language models into the input component, once again allowing tying between the embedding and softmax. This fills a significant gap in the language modeling architecture space, and the perplexity results bear out the advantages of combining adaptively-sized representations with weight tying. While the advance is in some sense fairly incremental, the centrality of unsupervised language modeling to modern deep NLP (ELMo, BERT, etc.) implies that perplexity improvements as large as this one may have meaningful downstream effects on performance on other tasks. Some things I noticed:

- One comparison that I believe is missing (I could be misreading the tables) is comparing directly to Merity et al.'s approach (adaptive softmax but fixed embedding/softmax dimension among the bands). Presumably you're faster, but is there a perplexity trade-off?

- The discussion/explanation of the differing performance of tying or not tying each part of the embedding weights for the different datasets is confusing; I think it could benefit from tightening up the wording but mostly I just had to read it a couple times. Perhaps all that's complicated is the distinction between embedding and projection weights; it would definitely be helpful to be as explicit about that as possible upfront.

- The loss by frequency-bin plots are really fantastic. You could also try a scatterplot of log freq vs. average loss by individual word/BPE token.

- Do you have thoughts as to why full-softmax BPE is worse than adaptive softmax word level? That goes against the current (industry) conventional wisdom in machine translation and large-scale language modeling that BPE is solidly better than word-level approaches because it tackles the softmax bottleneck while also sharing morphological information between words.

---

> ### Author Response · Authors · 2018-11-20
> **Response to Reviewer #1**
>
> We thank the reviewer for the comments!
>
> Q: “comparing directly to Merity et al.'s approach”
> Merity et al. share the input and output embeddings via an adaptive softmax where all words have the same embedding size. We reimplemented their approach and found that it did not perform very well in our experiments (25.48 PPL; Appendix A, Table 6, last row). We found that sharing fixed size input and output embeddings for a flat softmax performs better (22.63 PPL; second to last row of Table 6). This is likely because we train all words at every time step, which is not the case for an adaptive softmax with fixed size embeddings.
>
> Q: “The discussion/explanation of the differing performance of tying or not tying each part of the embedding weights for the different datasets is confusing”
> We updated the paper and hope that the discussion is clearer now. Thank you for the feedback!
>
> Q: “thoughts as to why full-softmax BPE is worse than adaptive softmax word level”
> Full-softmax BPE is worse because we measure perplexity on the word-level. This involves multiplying the probabilities of the individual BPE tokens. BPE token-level perplexity itself is actually significantly lower than word-level PPL (around 21.5 for GBW and around 18 for WikiText-103 for the models presented in the paper) but the two are not comparable.

---

### Public Comment · (anonymous) · 2018-11-12
**Hello the source code link in your paper is unaccessible?**

Code and pre-trained models are available at http://anonymized.

it is not available, would you fix it and I am very interested in your paper.

---

> ### Author Response · Authors · 2018-11-20
> **open sourcing**
>
> We are planning to open source the code and pre-trained models in the future.

---

### Author Response · Authors · 2018-11-20
**Updated version of the paper**

We updated the paper with the following changes:
* Table 3 contains new (better) validation results for WikiText-103. Note that only the validation numbers are updated, the test results were not affected. As described in the paper, we form training examples by taking 512 contiguous words from the training data with no regard for sentence boundaries. Evaluation is the same except that we require blocks to contain complete sentences of up to 512 tokens. Previously reported validation numbers did not always contain complete sentences because samples were built the same way as during training. We have corrected this so that validation is conducted the same way as testing.
* We also added new (and better) Billion word results with a bigger model achieving 23.7 perplexity.
* We added a comparison to Merity et al. fixed size adaptive softmax to the Appendix (Table 6).
* Clarified discussion around tying and not tying projections/word embeddings.

---

### Meta-Review · Area_Chair1 · 2018-12-11
**clear consensus to accept**

**Confidence:** 5
**Recommendation:** Accept (Poster)

**Metareview:**

There is a clear consensus among the reviews to accept this submission thus I am recommending acceptance. The paper makes a clear, if modest, contribution to language modeling that is likely to be valuable to many other researchers.